# Assessing Ionizing Radiation and Chlorine Dioxide (ClO₂) as Potential Aseptization Treatments for Yeast Recycling on Mixed Wort of Corn and Sugarcane in Brazil

Rafael Douradinho [1,*], Pietro Sica [2], Matheus Oliveira [1], Alana Uchoa Pinto [1], Layna Mota [3], Eduardo Mattos [1], Danilo Perecin [4], Vanessa Garcilasso [4], João Monnerat Araujo Ribeiro de Almeida [5], Sonia Piedade [6], Lucílio Alves [7], Valter Arthur [3], Suani Coelho [4] and Antonio Baptista [1,*]

1   Department of Agri-Food Industry, Food and Nutrition, College of Agriculture "Luiz de Queiroz", University of São Paulo, Padua Dias Avenue, 11, Piracicaba 13148-900, SP, Brazil; mathribeiro@usp.br (M.O.); alanauchoap@usp.br (A.U.P.); eduardo.castro.mattos@usp.br (E.M.)
2   Department of Plant and Environmental Sciences, University of Copenhagen, Thorvaldsenvej, 40, 1821 Frederiksberg, Denmark; pietro@plen.ku.dk
3   Center for Nuclear Energy in Agriculture, University of São Paulo, Centenário Avenue, 303, Piracicaba 13416-000, SP, Brazil; layna.amorim@usp.br (L.M.); arthur@cena.usp.br (V.A.)
4   Institute of Energy and Environment, University of São Paulo, Prof. Luciano Gualberto Avenue, 1289, São Paulo 05508-900, SP, Brazil; daniloperecin@usp.br (D.P.); vpecora@iee.usp.br (V.G.); suani@iee.usp.br (S.C.)
5   Chemistry Institute, Federal University of Rio de Janeiro, Athos da Silveira Ramos Avenue, 149, Rio de Janeiro 21941-909, RJ, Brazil; j.monnerat@iq.ufrj.br
6   Department of Math, Chemistry and Statistics, College of Agriculture "Luiz de Queiroz", University of São Paulo, Padua Dias Avenue, 11, Piracicaba 13148-900, SP, Brazil; soniamsp@usp.br
7   Department of Economics, Administration and Sociology, College of Agriculture "Luiz de Queiroz", University of São Paulo, Padua Dias Avenue, 11, Piracicaba 13148-900, SP, Brazil; lralves@usp.br
*   Correspondence: rafael.douradinho@usp.br (R.D.); asbaptis@usp.br (A.B.)

**Abstract:** Yeast recycling, which is a common practice in sugarcane ethanol plants, could be expanded if it could be successfully implemented in corn-based ethanol production. However, the process of recycling the yeast remaining after fermentation is hampered by contaminating microorganisms that reduce the fermentation efficiency and compete with the yeast for the fermentable sugars. Currently, antibiotics are used to control microbial contamination. This study proposes chlorine dioxide and electron beam irradiation as alternative control methods for completely inactivating contaminants and minimizing their effect on recycled yeast. For that, wort sterilization using electron radiation (20 kGy) and treatment with a chemical biocide, namely chlorine dioxide (125 mg kg$^{-1}$), were compared with non-treated wort. Five fermentation cycles were performed using fed-batch systems with 300 g L$^{-1}$ of fermentable sugars. The results obtained in this study showed the inactivation of contaminants under the effect of electron beam irradiation, which led to an increase in the productivity, yield, and efficiency of fermentation by 0.21 g L$^{-1}$h$^{-1}$, 2.6%, and 4.7%, respectively. However, ClO₂ did not show promising results in reducing contamination or improving fermentative parameters. Thus, electron beam irradiation of contaminated wort may be a suitable alternative to chemical biocides and would allow the use of recycled yeast in corn-based ethanol plants.

**Keywords:** bioenergy; biofuel; corn ethanol; fermentation; yeast recycling; microbial contamination; disinfection; sterilization; ionizing radiation; electron beam

## 1. Introduction

Traditionally, the United States produces ethanol primarily from corn [1], while Brazil uses sugarcane as the main feedstock for ethanol production [2]. In recent years, the corn ethanol industry in Brazil has been growing steadily [3]. The Brazilian National Biofuels Policy (RenovaBio), implemented in 2017 (Law No. 13.576/2017), has encouraged

producers to invest in the efficiency and sustainability of the production processes; it provides a financial credit to biofuel producers, the Decarbonization Credit (CBIO). One CBIO is equivalent to one ton of $CO_2$ emissions avoided, and its price ranges from USD 20 to USD 40 [4]. Brazil's commitments made at the 2015 Climate Change Conference (COP21) also require more efficient and sustainable bioenergy production in the national energy matrix. The substitution of fossil fuels with biofuels is also in line with the 2030 Agenda for Sustainable Development, adopted by all United Nations member states at COP21, which is aligned with Sustainable Development Goals (SDGs) 7 and 13, affordable and clean energy and climate action, respectively [5].

In this context, the implementation of corn ethanol industries in Brazil requires solutions to properly adapt the traditional American production process. These adaptations include the integration with sugarcane or energy cane to provide fiber to meet the industry's energy demand and using the cane juice, which serves as a source of fermentable sugars and nutrients, to dilute the corn [6,7]. However, microbial contamination is one of the most significant factors affecting fermentation efficiency [8].

Contaminating bacteria can either produce compounds that inhibit the fermentation process or compete with yeasts for sugars and nutrients in the wort [9] and can also have a deleterious effect on the yeast, reducing fermentation efficiency [10]. Contaminants are typically transported together with the raw materials (corn [9] and sugarcane [11,12]) from the field to the agro-industry.

Currently, antibiotics are used to control bacterial infections in ethanol distilleries. The widespread use of certain antibiotics may promote bacterial antibiotic resistance [13]. Moreover, they do not prevent reinfection by *Lactobacillus* spp., which can form biofilms on the internal surfaces of industrial pipes, becoming even more tolerant to high concentrations of antibiotics [14,15]. In addition, some antibiotics, such as virginiamycin, persist in yeast and distiller's dried grains (DDGs), resulting in undesired residual effects on high-value animal feed products [16].

Brazilian distilleries and recent studies have been seeking alternatives to antibiotics for wort disinfection. Most of the known methods for inactivating microorganisms require further adjustment to suit their sustainability and overcome any limitation for a prompt application on a large scale as a wort treatment for distilleries. The use of steam is demanding in terms of time and energy [17]. Gamma radiation also requires too long of a time for treatment due to the low dose rate of cobalt-60 [18], and natural bacterial antagonists have been studied as bacteriophages [19].

Chemical compounds with bacteriostatic and bactericidal activities have also been tested. One alternative is the use of biocides, such as chlorine dioxide, which is known for its high antimicrobial efficacy [20]. Another option is the use of an electron beam as ionizing radiation [2,21]. It has the potential to treat worts for different types of fermentation, as ionizing radiation has also been applied for fermentative hydrogen production, for example [22]. Industrial ethanol production processes do not require ideal conditions, such as complete inactivation of contaminating microorganisms in the wort, and the fermentation does not have to be under aseptic conditions [10,23–25]. However, the contamination limits improvements to the current processes, fermentation efficiency, and industry revenues.

In this study, we propose the adoption of the Melle-Boinot yeast recycling process [26] in the corn ethanol production system. The Melle-Boinot fermentation process is used in 83% of the first-generation ethanol mills in Brazil [27]. It consists of fed-batch fermentations with cell recycling [28]. This is a strategy for avoiding osmotic stress on yeasts [6] and also avoiding the consumption of sugars for yeast growth and budding in each fermentation cycle [29].

However, the negative effects of microbial contamination and the metabolites formed by it tend to increase after each yeast recycling cycle [15,16], making yeast recovery challenging in some cases [26,27]. Therefore, alternative processes for replacing the use of antibiotics for wort disinfection and possibly achieving wort sterilization merit research aiming to enable the adoption of the Melle-Boinot process in corn ethanol production.

Based on this background, the main objective of this study was to assess how chlorine dioxide ($ClO_2$) and ionizing radiation treatments would affect fermentative parameters during five cycles of yeast recycling in corn-based wort with sugarcane. Our hypothesis was that for yeast recycling in mixed wort fermentation to be viable, efficient aseptization treatments are needed to reduce the cumulative negative effects of contaminants and their metabolites throughout the cycles.

## 2. Results and Discussion

### 2.1. Treatment Efficiency in Contamination Control and Acidity

The treatments had no significant effect on the counts of mesophilic microorganisms at the end of all fermentation cycles since they represent the yeasts added in the wort (from 8.3 to 8.6 Log CFU + 1 $mL^{-1}$). The electron beam irradiation was efficient in wort sterilization (0 Log CFU + 1 $mL^{-1}$). However, T1 had minimal recontamination, as indicated by the results other than zero for total bacteria counts on wine (0.63–1.3 Log CFU + 1 $mL^{-1}$) (Table 1).

**Table 1.** Mesophilic and total bacterial contamination in the wine of the five fermentation cycles with yeast recycling performed in this study for wort disinfected with electron beam (radiation), wort disinfected with chlorine dioxide ($ClO_2$), and non-treated wort (No treat.).

| Cycle | Treatment | Total Mesophilic [n.s.] | Total Bacteria |
|---|---|---|---|
| | | Log (CFU + 1) $mL^{-1}$ | Log (CFU + 1) $mL^{-1}$ |
| 1 | Radiation | $8.5 \pm 0.3$ | $0.63 \pm 0.9$ Ba |
| | $ClO_2$ | $8.3 \pm 0.1$ | $4.1 \pm 0.1$ Ab |
| | No treat. | $8.6 \pm 0.1$ | $5.4 \pm 1.6$ Aa |
| 2 | Radiation | $8.4 \pm 0.2$ | $0.97 \pm 1.6$ Ba |
| | $ClO_2$ | $8.3 \pm 0.1$ | $5.6 \pm 2.3$ Aab |
| | No treat. | $8.4 \pm 0.1$ | $6.1 \pm 2.4$ Aa |
| 3 | Radiation | $8.4 \pm 0.1$ | $1.3 \pm 1.4$ Ba |
| | $ClO_2$ | $8.5 \pm 0.2$ | $5.7 \pm 2.2$ Aab |
| | No treat. | $8.5 \pm 0.2$ | $6.3 \pm 2.1$ Aa |
| 4 | Radiation | $8.3 \pm 0.2$ | $1.3 \pm 1.5$ Ba |
| | $ClO_2$ | $8.4 \pm 0.3$ | $7.8 \pm 0.2$ Aa |
| | No treat. | $8.4 \pm 0.2$ | $8.2 \pm 0.1$ Aa |
| 5 | Radiation | $8.4 \pm 0.2$ | $1.3 \pm 1.4$ Ba |
| | $ClO_2$ | $8.3 \pm 0.2$ | $5.9 \pm 1.3$ Aab |
| | No treat. | $8.4 \pm 0.1$ | $7.9 \pm 0.5$ Aa |
| Average | Radiation | $8.4 \pm 0.2$ | $1.1 \pm 1.4$ Ba |
| | $ClO_2$ | $8.4 \pm 0.2$ | $5.8 \pm 1.2$ Aab |
| | No treat. | $8.5 \pm 0.1$ | $6.8 \pm 1.3$ Aa |

Standard errors are indicated after the mean. Different uppercase letters indicate a significant difference between treatments within the same cycle (Tukey < 0.05); different lowercase letters indicate a significant difference for the same treatment in different cycles. [n.s.] indicates that there was not a significant difference between treatments and among the cycles in the wine mesophilic contamination.

The chlorine dioxide treatment resulted in a lower total acidity in the wine of cycles 2, 4, and 5 compared to the ionizing radiation treatment. In terms of pH, the chlorine dioxide treatment also resulted in higher values in the wine of cycles 2, 3, 4, and 5 compared to the ionizing radiation treatment (Table 2).

**Table 2.** Total acidity (titratable acidity) and pH in the wine of the five fermentation cycles with yeast recycling performed in this study for the wort disinfected with electron beam (radiation), wort disinfected with chlorine dioxide ($ClO_2$), and non-treated wort (No treat.).

| Cycle | Treatment | Total Acidity (g $L^{-1}$) | pH |
|---|---|---|---|
| 1 | Radiation | 6.2 ± 0.6 Aa | 4.50 ± 0.01 Aa |
| | $ClO_2$ | 5.7 ± 0.4 Aa | 4.59 ± 0.11 Aa |
| | No treat. | 6.2 ± 0.7 Aa | 4.52 ± 0.03 Aa |
| 2 | Radiation | 5.3 ± 0.2 Ab | 4.47 ± 0.02 Bab |
| | $ClO_2$ | 4.5 ± 0.3 Bb | 4.54 ± 0.02 Aa |
| | No treat. | 4.8 ± 0.3 Bb | 4.47 ± 0.05 Ba |
| 3 | Radiation | 4.5 ± 0.4 Ac | 4.52 ± 0.03 Ba |
| | $ClO_2$ | 4.1 ± 0.3 Ab | 4.64 ± 0.08 Aa |
| | No treat. | 4.5 ± 0.1 Ab | 4.55 ± 0.08 ABa |
| 4 | Radiation | 4.8 ± 0.2 Abc | 4.44 ± 0.03 Bb |
| | $ClO_2$ | 4.0 ± 0.3 Bb | 4.64 ± 0.10 Aa |
| | No treat. | 4.5 ± 0.1 Ab | 4.47 ± 0.08 Ba |
| 5 | Radiation | 4.7 ± 0.2 Abc | 4.51 ± 0.03 Ba |
| | $ClO_2$ | 4.2 ± 0.1 Bb | 4.65 ± 0.08 Aa |
| | No treat. | 4.6 ± 0.1 Ab | 4.49 ± 0.08 Ba |
| Average | Radiation | 5.1 ± 0.3 Ab | 4.49 ± 0.02 Ba |
| | $ClO_2$ | 4.5 ± 0.3 Bb | 4.61 ± 0.08 Aa |
| | No treat. | 4.9 ± 0.3 Bb | 4.50 ± 0.06 Ba |

Standard errors are indicated after the mean. Different uppercase letters indicate a significant difference between treatments within the same cycle (Tukey < 0.05); different lowercase letters indicate a significant difference for the same treatment in different cycles.

### 2.1.1. Chlorine Dioxide

Chlorine dioxide treatment is effective in controlling antibiotic-resistant bacterial strains [15,16,30] and does not leave residues in the yeast cells, which is a critical requirement when the biomass (WDG or DDG) is intended for animal nutrition [31]. Meneghin et al. (2008) observed that at a dose of 50 mg $L^{-1}$, the antimicrobial treatment affected the viability of industrial strains of *S. cerevisiae*, and at 100 mg $L^{-1}$, it completely inhibited yeast growth [30]. Additionally, Zhu, Chen, and Yu (2013) noted DNA damage at a concentration of 100 mg $L^{-1}$ [32]. In this study, chlorine dioxide at a dose of 125 mg $kg^{-1}$ was added directly to the wort as a way to gradually make it available (fed-batch) to the yeast cream (indirect). This approach was intended to minimize potential stress on the *S. cerevisiae* cells [6,30].

However, it is worth noting that this treatment significantly reduced the yield compared to the negative control and exhibited lower productivity and efficiency compared to ionizing radiation (see Table 3). As a strong oxidizing agent, $ClO_2$ had an undesirable effect on yeast performance.

Furthermore, our results suggest that the gradual application of chlorine dioxide did not have a positive impact on the control of microbial contamination. Thus, although $ClO_2$ biocide treatment of the wort with an amount of 125 mg $kg^{-1}$ is cheaper (~3 USD $m^{-3}$) than the energy cost of an electron accelerator producing a dose of 20 kGy (4 USD $m^{-3}$) [33,34], this study showed that $ClO_2$ treatment was less effective than electron irradiation in suppressing microbial contamination.

**Table 3.** Residual sugars, productivity, fermentation efficiency, and yield of the five fermentation cycles with yeast recycling performed in this study for wort disinfected with electron beam (radiation), wort disinfected with chlorine dioxide ($ClO_2$), and non-treated wort (No treat.).

| Cycle | Treatment | Residual Sugars | Productivity | Efficiency | Yield |
|---|---|---|---|---|---|
| | | $g\ L^{-1}$ | $g\ L^{-1}\ h^{-1}$ | % | % |
| 1 | Radiation | 87.4 ± 3.3 Aa | 3.11 ± 0.0 Ad | 96.2 ± 1.4 Aa | 67.4 ± 1.1 Ac |
| | $ClO_2$ | 88.0 ± 12.8 Aa | 2.90 ± 0.2 Ba | 93.4 ± 0.8 Ba | 64.3 ± 0.6 Aa |
| | No treat. | 82.3 ± 18.7 Aa | 3.00 ± 0.1 Aba | 93.3 ± 0.3 Ba | 66.6 ± 0.2 Aa |
| 2 | Radiation | 79.1 ± 5.1 Aa | 3.28 ± 0.0 Abcd | 97.2 ± 1.4 Aa | 70.6 ± 1.1 Abc |
| | $ClO_2$ | 76.0 ± 5.8 Aba | 3.03 ± 0.3 Aa | 92.0 ± 3.1 Ba | 70.3 ± 2.5 Aa |
| | No treat. | 62.8 ± 12.3 Ba | 3.29 ± 0.1 Aa | 93.5 ± 1.4 Ba | 73.8 ± 1.1 Aa |
| 3 | Radiation | 43.4 ± 17 Ac | 3.81 ± 0.3 Aa | 96.8 ± 0.7 Aa | 82.7 ± 0.6 Aa |
| | $ClO_2$ | 57.3 ± 20.2 Aa | 3.22 ± 0.6 Ba | 91.1 ± 5.4 Ba | 72.5 ± 4.3 Aa |
| | No treat. | 50.9 ± 10.1 Aa | 3.35 ± 0.3 Ba | 91.8 ± 3.1 Ba | 75.5 ± 2.5 Aa |
| 4 | Radiation | 81.1 ± 10.7 Aa | 3.24 ± 0.2 Acd | 97.3 ± 0.9 Aa | 70.3 ± 0.7 Abc |
| | $ClO_2$ | 70.2 ± 33.6 Aa | 3.03 ± 0.4 Aa | 91.3 ± 2.9 Ba | 68.0 ± 2.3 Aa |
| | No treat. | 72.2 ± 34.7 Aa | 2.99 ± 0.5 Aa | 89.9 ± 2.3 Ba | 67.0 ± 1.8 Aa |
| 5 | Radiation | 55.2 ± 9.7 Ab | 3.56 ± 0.2 Aab | 96.1 ± 1.9 Aa | 77.2 ± 1.5 Aab |
| | $ClO_2$ | 81.1 ± 21.7 Aa | 2.83 ± 0.4 Ba | 86.9 ± 2.7 Ca | 61.9 ± 2.2 Ba |
| | No treat. | 59. ± 27.2 Aa | 3.31 ± 0.3 Aba | 91.3 ± 1.1 Ba | 72.3 ± 0.9 Aba |
| Average | Radiation | 69.2 ± 18.9 Aab | 3.40 ± 0.3 Abc | 96.7 ± 0.6 Aa | 73.6 ± 6.2 Abc |
| | $ClO_2$ | 74.5 ± 11.6 Aa | 3.00 ± 0.1 Ba | 90.9 ± 2.4 Ba | 67.4 ± 4.3 Ba |
| | No treat. | 65.4 ± 12.1 Aa | 3.19 ± 0.2 Ba | 92.0 ± 1.5 Ba | 71.0 ± 4.0 Aba |

Standard errors are indicated after the mean. Different uppercase letters indicate a significant difference between treatments within the same cycle (Tukey < 0.05); different lowercase letters indicate a significant difference for the same treatment in different cycles.

### 2.1.2. Ionizing Radiation

Our results, based on colony-forming units, demonstrate that ionizing radiation is effective in inactivating microorganisms. In this study, we ensured the sterilization of all equipment and apparatuses using an autoclave to prevent external contamination. This practice played a crucial role in maintaining the conditions and effects established by the treatments. The counting results found at the end of the fermentation cycles for the ionizing radiation treatment showed levels insufficient to express detrimental effects on the fermentation process [35–37]. Chlorine dioxide did not show a significant difference from the negative control in any of the five cycles.

Alcarde, Walder, and Horii (2001), quoting [38], report improved fermentation efficiency in worts irradiated with doses as low as 0.6 kGy [18]. However, higher irradiation levels are required for the complete inactivation of contaminating microorganisms [39]. Douradinho et al. (2023), for instance, achieved complete inactivation with 20 kGy [6], and a significant reduction in organic acid production was observed by the authors of [8]. Therefore, our results demonstrate the potential of a 20 kGy electron beam in controlling microorganisms and improving fermentation parameters. The ionizing ability of the electron beam to sterilize the wort was immediate (~1 s) and did not damage the yeast.

### 2.2. Effects of Treatments on Yeast Viability and Reproduction

The ionizing radiation treatment (T1) had a significantly higher initial yeast viability in cycle 3, compared to the other treatments. In the second cycle, T1 also had significantly higher sprouting rates than the negative control at the beginning of the fermentation. And in terms of the yeast population, the ionizing radiation treatment resulted in significantly higher values at the end of cycle 2 and at the beginning of cycle 3 (Table 4).

**Table 4.** The initial and final yeast cell viability, sprouting rates, and population of the five fermentation cycles with yeast recycling performed in this study for wort disinfected with ionizing radiation (radiation), wort disinfected with chlorine dioxide (ClO$_2$), and non-treated wort (No treat.).

| Cycle | Treatment | Yeast Cell Viability | | Sprouting Rates | | Population | |
| --- | --- | --- | --- | --- | --- | --- | --- |
| | | % | | % | | Log (cel. + 1) mL$^{-1}$ | |
| | | Initial | Final | Initial | Final | Initial | Final |
| 1 | Radiation | 85.9 ± 0 Aa | 44.1 ± 3 Aa | 3.20 ± 3.8 Aa | 1.27 ± 0.9 Abc | 9.27 ± 0.2 Aa | 9.02 ± 0.2 Aa |
| | ClO$_2$ | 84.1 ± 0 Aa | 34.8 ± 10 Aa | 3.17 ± 3.8 Aa | 0.93 ± 0.3 Aa | 9.23 ± 0.1 Aa | 8.88 ± 0.2 Aa |
| | No treat. | 84.7 ± 0 Aa | 31.9 ± 17 Aa | 3.10 ± 3.7 Aab | 0.06 ± 0.1 Bc | 9.22 ± 0.1 Aa | 8.69 ± 0.6 Aa |
| 2 | Radiation | 45.4 ± 2 Ab | 36.2 ± 8 Aa | 1.25 ± 1.1 Aa | 0.57 ± 0.4 Ac | 9.04 ± 0.2 Aab | 8.90 ± 0.2 Aa |
| | ClO$_2$ | 32.5 ± 10 Ab | 25.9 ± 2 ABab | 0.95 ± 0.3 ABa | 1.88 ± 1.6 Aa | 8.83 ± 0.2 Ab | 8.65 ± 0.1 Bab |
| | No treat. | 26.2 ± 20 Ab | 22.7 ± 6 Ba | 0.10 ± 0.1 Bb | 0.90 ± 0.9 Abc | 8.54 ± 0.6 Ab | 8.70 ± 0.1 Ba |
| 3 | Radiation | 35.9 ± 7 Ac | 22.3 ± 2 Ab | 0.50 ± 0.3 Aa | 2.46 ± 0.7 Aabc | 8.87 ± 0.2 Ab | 8.55 ± 0.1 Ab |
| | ClO$_2$ | 25.9 ± 2 Bb | 22.9 ± 4 Ab | 1.88 ± 1.6 Aa | 3.88 ± 3.3 Aa | 8.65 ± 0.1 Bb | 8.64 ± 0.3 Aab |
| | No treat. | 22.7 ± 6 Bb | 20.8 ± 4 Aa | 0.90 ± 0.9 Ab | 3.13 ± 1.8 Aab | 8.70 ± 0.1 ABab | 8.57 ± 0.2 Aa |
| 4 | Radiation | 22.3 ± 2 Ad | 23.2 ± 3 Ab | 2.46 ± 0.7 Aa | 4.49 ± 2.4 Aab | 8.55 ± 0.1 Ac | 8.47 ± 0.1 Ab |
| | ClO$_2$ | 24.2 ± 3 Ab | 20.4 ± 6 Ab | 4.54 ± 3.4 Aa | 4.38 ± 1.2 Aa | 8.64 ± 0.3 Ab | 8.45 ± 0.1 Ab |
| | No treat. | 20.8 ± 5 Ab | 20.4 ± 4 Aa | 3.56 ± 1.7 Aab | 5.14 ± 2.4 Aa | 8.66 ± 0.1 Aab | 8.62 ± 0.1 Aa |
| 5 | Radiation | 23.2 ± 3 Abd | 18.6 ± 3 Ab | 4.49 ± 2.4 Aa | 5.26 ± 3.1 Aa | 8.47 ± 0.1 Ac | 8.46 ± 0.1 Ab |
| | ClO$_2$ | 21.4 ± 6 Ab | 17.6 ± 1 Ab | 4.38 ± 1.2 Aa | 4.76 ± 1.5 Aa | 8.54 ± 0.2 Ab | 8.44 ± 0.1 Ab |
| | No treat. | 19.0 ± 5 Ab | 16.7 ± 3 Aa | 5.14 ± 2.4 Aa | 4.95 ± 1.3 Aa | 8.60 ± 0.1 Aab | 8.57 ± 0.1 Aa |
| Average | Radiation | 42.5 ± 3 Ab | 28.9 ± 4 Aab | 2.38 ± 1.7 Aa | 2.81 ± 1.5 Aabc | 8.84 ± 0.2 Ab | 8.68 ± 0.1 Aab |
| | ClO$_2$ | 37.6 ± 4 Ab | 24.3 ± 5 Aab | 2.98 ± 2.1 Aa | 3.17 ± 1.6 Aa | 8.78 ± 0.2 Ab | 8.61 ± 0.2 Aab |
| | No treat. | 34.7 ± 7 Ab | 22.5 ± 7 Aa | 2.56 ± 1.8 Aab | 2.84 ± 1.3 Aab | 8.74 ± 0.2 Aab | 8.63 ± 0.2 Aa |

Standard errors are indicated after the mean. Different uppercase letters indicate a significant difference between treatments within the same cycle (Tukey < 0.05); different lowercase letters indicate a significant difference for the same treatment in different cycles.

### 2.3. Effects of Contamination on Acidity and Yeast Viability

One of the main challenges for the integration of sugarcane into the corn ethanol production process is the high contaminant load brought from the field [7,9,12]. Li et al. (2016) demonstrated a higher bacterial diversity in the corn ethanol production process [40]. While bacteria of the genus Lactobacillus were predominant in two of the five industrial units studied, the other three production units showed a predominance of *Pseudomonas*, *Escherichia*, and *Shigella* [40]. The co-products of these contaminating bacteria can significantly impact the metabolism and development of yeasts [41]. These compounds are mainly represented by organic acids [42,43], diacetyl, hydroxylated fatty acids [23], reuterin, and gummy biofilms [15,20,44,45].

In this study, a comparison of the treatments revealed that the non-treated wort and wort treated with chlorine dioxide had higher contamination rates. However, these treatments did not have higher acidity and lower pH compared to the ionizing treatment, in which the microorganisms were inactivated. A decrease in pH was observed in all treatments, comparing wort (4.9) and wine (4.4–4.6) (Table 2). Thus, the acidification of the medium and reduction in its pH throughout the process can be attributed to the fermentation agent itself [21,27,46,47]; after the ionizing radiation treatment, the bacteria and wild yeasts were completely inactivated.

The excretion of some acids by the yeast may be a consequence of the metabolism of the synthesis of fatty acids, steroids, and amino acids [48]. These acids produced by yeast also have the function of inhibiting bacterial growth, a condition that may have contributed to the limitation of the development of contaminants in the non-treated wort and wort treated with chlorine dioxide [49,50].

A considerable increase in succinic acid during fermentation was found by Silva et al. (2023) even in the absence of contaminating microorganisms [8]. Yeast primarily produces succinic acid, which can reach concentrations of up to 1.7 g L$^{-1}$ [50,51]. This suggests that succinate synthesis is primarily yeast-derived and influenced by contaminants. Although there is no known physiological purpose for the large amount of succinic acid excreted by

yeast, it serves an ecological role by increasing the yeast's competitiveness in industrial fermentation environments [50].

In this study, contamination did not significantly affect the initial and final yeast cell viability and population in most of the cycles. These findings are consistent with those reported by Cherubin (2003) [52]. In a comparative experiment involving *S. cerevisiae* strains in pure culture and mixed culture with *L. fermentum* bacteria during six fermentative cycles, the author observed no correlation between bacterial contamination and yeast viability. However, in fifteen-cycle fermentations with yeast recycling, Oliva-Neto et al. (1994) found a 64% reduction in yeast cell viability in the presence of contaminants [53].

*2.4. Fermentative Parameters*

The ethanol content in the wine was significantly higher for the wort treated with ionizing radiation compared to the chlorine dioxide treatment in cycles 1, 3, and 5. In cycle 3, the wort treated with ionizing radiation also had a higher ethanol content than the non-treated wort. Considering the average ethanol content of all cycles, the ionizing radiation treatment resulted in the significantly highest content compared to the other two treatments (Figure 1).

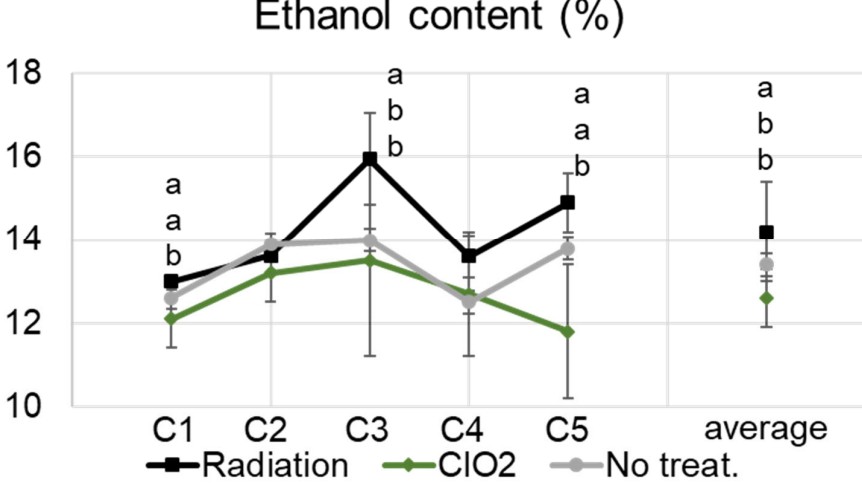

**Figure 1.** Ethanol content in the wine at the end of the five cycles (C) for wort treated with ionizing radiation (radiation), wort treated with ionizing radiation chlorine dioxide ($ClO_2$), and non-treated wort (No treat.). The average of all cycles for each treatment is also represented. Bars indicate standard deviation. Different letters indicate a significant difference between treatments.

The treatments assessed in this study had little effect on the residual sugars. The only significant difference among treatments was observed in cycle 2, in which the wort subjected to irradiation treatment had a higher residual sugar content than the non-treated wort. In cycles 1, 3, and 5, the ionizing radiation treatment resulted in higher values of productivity than the chlorine dioxide treatment. In terms of efficiency, the radiation treatment also resulted in significantly higher values in all the cycles. In cycle 5, the negative control resulted in a higher efficiency than the chlorine dioxide treatment. Chlorine dioxide also resulted in a lower average yield compared to the other two treatments (Table 3).

*2.5. Effects of Contamination on Fermentative Parameters*

Although we did not observe population growth of contaminating microorganisms throughout the fermentation cycles, the impact of these contamination levels on the productivity, yield, and efficiency of alcoholic fermentation was significant ($p < 0.05$). In the average of the five cycles, these parameters showed a reduction of at least 0.21 g $L^{-1}h^{-1}$, 2.6%, and 4.7%, respectively. It is worth noting that the reduction in alcoholic fermentation yield due to contaminating microorganisms is a phenomenon observed by other researchers as well.

For example, in a 30-h fermentation with a population of $4.5 \times 10^8$ CFU mL$^{-1}$, the authors of [54] observed a yield reduction of approximately 17%. Similarly, the authors of [35] reported a yield reduction of up to 55% when fermentations were contaminated with more than $10^7$ bacteria cells per mL. In another study, Khan et al. (1990) noted an 11.4% lower yield in fermentations with contamination levels around 6.7 Log (CFU + 1) mL$^{-1}$ [37].

In the five-cycle average, the alcoholic fermentation efficiency and productivity observed in wort irradiated with a 20 kGy electron beam were, respectively, 6.4% and 13.3% higher ($p < 0.05$) than those observed in the chlorine dioxide treatment. In addition, they were 5.1% and 6.6% higher ($p < 0.05$) than those observed in the negative control.

In fermentations involving yeast recycling over 15 cycles, Oliva-Neto et al. (1994) noted a significant 26-percentage-point reduction in alcoholic fermentation yield when fermentation was conducted in the presence of contaminants [53]. Bischoff et al. (2009) reported a 27% reduction in the yield of alcoholic fermentation in wort obtained from corn saccharification and contaminated with *Lactobacillus fermentum* at $10^8$ CFU mL$^{-1}$ [55], which was corroborated by the range reported by Brexó and Sant'Ana (2017) [20].

While it is expected that the proliferation of bacteria and the level of contamination would evolve throughout the fermentation cycles [10,20,27] in worts without complete contaminant inactivation, both the chlorine dioxide and negative control wines did not exceed a bacterial count higher than 8.21 Log (CFU + 1) mL$^{-1}$. This limitation on bacterial development can be attributed to the bactericidal effect of ethanol itself, produced by the yeasts [50]. Thomas et al. (2001) also noted bacterial growth inhibition when yeasts and contaminating bacteria were inoculated simultaneously, both at a population level of 7 Log (CFU + 1) mL$^{-1}$ [56]. Additionally, Oliva-Neto et al. (2004) and Meneghin et al. (2010) reported the possibility of antibacterial activity by some industrial strains of *S. cerevisiae* [57,58]. Ethanol losses of approximately 1.5% were observed in industrial corn fermentations contaminated with $10^9$ bacteria mL$^{-1}$ [41], and losses ranging from 1% to 3% occurred in fermentations with a bacterial population on the order of $10^8$ cells mL$^{-1}$ [59].

### 2.6. Effects of Wort Composition on Very High Gravity Fermentation

In addition to metabolites, competition for nutrients between contaminating microorganisms and yeast can significantly impact fermentation performance [44]. The wort used in this study, obtained from the hydrolysis of corn in sugarcane juice, was nutritionally deficient in essential elements: nitrogen (885.8 mg L$^{-1}$) and phosphorus (444.4 mg L$^{-1}$) were below the minimum concentrations required for very high gravity fermentation: N (1000 mg L$^{-1}$) and P (1000 mg L$^{-1}$) (Table 5) [60].

**Table 5.** Mixed wort composition.

|  | Concentration (mg L$^{-1}$) | Minimum Required [a] (mg L$^{-1}$) |
|---|---|---|
| Fermentable sugars | 342.40 ± 2.2 | - |
| Potassium | 636.6 ± 48.5 | 117 |
| Phosphorus | 444.4 ± 14.0 | >1000 |
| Magnesium | 129.2 ± 23.3 | 72 |
| Sodium | 71.1 ± 24.9 | - |
| Zinc | 1.73 ± 1.1 | 0.39 |
| Copper | 2.93 ± 1.2 | 0.10 |
| Cobalt | 2.43 ± 1.0 | - |
| Manganese | 2.08 ± 0.1 | 0.16 |
| Total nitrogen | 885.8 ± 88.0 | >1000 |

[a] Minimum required contents for very high gravity fermentation, according to Walker (1998) [60].

This nutritional limitation had a pronounced effect on the experiments, leading to a significant reduction in yeast cell viability and fermentation inhibition [6,61]. Both phosphorus and nitrogen are essential for yeast activity and alcoholic fermentation [62,63] and directly affect cell viability during alcoholic fermentation [64]. Phosphorus constitutes an inorganic

component of yeast cells, accounting for between 0.8% and 2.6% of their dry weight [65]. It is required for energy metabolism (ATP production in glycolysis and the respiratory chain) and the synthesis of nucleic acids [66]. Nitrogen is also essential for both cell multiplication and fermentation, playing a role in the synthesis of protein biomolecules, nucleic acids, peptides, polyamides, and vitamins [67]. Nitrogen accounts for approximately 10% of the dry mass of yeast cells [68]. These elements are integral cellular components [65,68]. Thus, recovering yeast cells can provide nutritional benefits to subsequent cycles by transferring cellular components [53] such as amino acids, peptides, proteins, nucleic acids, and their degradation products [68], among other nutritional sources.

Douradinho et al. (2023) highlighted the critical role of supplementation with nutrients, such as phosphates found in sugarcane juice, in improving yeast tolerance to ethanol due to the nutritional enrichment of the wort [6]. Similarly, Sica et al. (2021) found that combining energy cane juice with corn also improved fermentation efficiency compared to using only corn [7]. These improvements resulted in a significant increase in ethanol content. However, they emphasize the importance of nutrient supplementation in achieving high ethanol levels in worts obtained from corn hydrolyzed even in cane juice [6,7].

Yeast tolerance to ethanol is intricately connected to the availability of nutrients in the wort. Yeast cells exhibit increased nutrient requirements under stressful conditions, and microelements, including mineral ions, are essential for maintaining pH and osmotic stability, facilitating nutrient transport, and serving as cofactors in fermentation reactions [69]. Mixed wort can increase by about 5% in fermentation efficiency compared to wort with just corn, mainly due to the nitrogen and phosphorus provided by the sugarcane juice, allowing the fermentation to reach higher ethanol contents and lower residual sugars [6,7].

The concentration of ethanol in the medium can exacerbate the nutritional challenges faced by yeast cells, thereby directly affecting the fermentation performance. It is worth noting that in this study, we supplemented the wort with 600 mg of urea per liter of wort. However, in our study, we opted not to supplement the wort with phosphorus sources, which could be a mechanism for increasing yeast tolerance to high ethanol levels. Thus, phosphorus supplementation could have resulted in increased residual sugar consumption, resulting in higher ethanol content. This also may explain why the maximum ethanol content we achieved in this study was approximately 16%, accompanied by significantly high residual sugar levels and yields lower than desirable (85%) (Figure 1 and Table 3).

### 2.7. Impacts in the Industrial Processes

In this study, the integral recycling of yeast cream was performed with the aim of assessing the reduction in the demand for new yeast inocula and minimizing the consumption of sugars for cell propagation. It is important to note that in this experiment, the acid treatment [70] was not necessary once the population of contaminating microorganisms was completely inactivated (0.00 Log (CFU + 1) mL$^{-1}$) in wort irradiated with a 20 kGy electron beam [6]. The electron beam was then effective in reducing the yeast recycling time since it was possible to eliminate the residence time of the yeast cells in an acidic medium. This approach, in turn, leads to improved overall yield and productivity of the process.

The control of bacteria in alcoholic fermentation remains a significant challenge in the industrial process. The inactivation of microorganisms in wort using electron beam irradiation (20 kGy) showed a positive effect on alcoholic fermentation. Considering a standard distillery producing 200,000 cubic meters of ethanol per year, based on the results of this study, we can estimate an increase of at least 10 million liters of ethanol per year just by inactivating contaminating microorganisms using electron beam irradiation, without requiring additional raw materials for the process.

The use or combination of the biocide chlorine dioxide warrants further in-depth research to determine its potential beneficial effects on the microbiota and alcoholic fermentation performance. This approach should aim to improve yeast metabolic activity and address industrial challenges. Another factor to be considered is related to the scale-up challenges and the economic feasibility of adopting ionizing radiation technology on an

industrial scale. In this study, the proposed system showed promise at the laboratory scale. Therefore, a pilot-scale experiment and an economic and technical assessment are further steps necessary for the adoption of this technology.

Another critical aspect to consider is that by inactivating the contaminating microorganisms present in the wort, it becomes possible to redefine the standard criteria for yeast selection [71–73]. Currently, yeast selection criteria focus primarily on the yeast's ability to survive under different biotic conditions (aggressiveness) [72]. Instead, the focus could be shifted to the selection of yeasts that show high performance under challenging abiotic conditions, i.e., efficiency and tolerance behavior.

## 3. Materials and Methods

All the experiments were conducted in the Sugar and Alcohol Laboratory of the Department of Agri-Food Industry, Food, and Nutrition of the College of Agriculture "Luiz de Queiroz", at the University of São Paulo (ESALQ/USP) campus, in Piracicaba. The wort irradiation was conducted at the Laboratory of Intense Radiation Sources (LFIR)—Electron Accelerator of the Center for Radiation Technology (CTR) at the Nuclear and Energetic Research Institute from the University of São Paulo.

The mixed wort was prepared by diluting the corn with sugarcane juice before the starch hydrolysis. After the hydrolysis, the wort was centrifuged, and the supernatant was filtered (<210 μm) to remove suspended solids, allowing for yeast recycling. The wort was contaminated and divided into three equal parts (treatments): (1) treated with electron beam as ionizing radiation (20 kGy); (2) treated with chlorine dioxide (125 mg kg$^{-1}$); (3) remained contaminated (untreated). The yeast was recycled for five cycles. In each cycle, we assessed mesophilic and total bacterial contamination, total acidity, and pH in the wine. We also evaluated the initial and residual fermentable sugars, the final ethanol content, yeast viability, sprouting rates, and the yeast population at the beginning and end of each fermentation cycle. The experiments and processes described in this study were conducted in February and March 2022.

### 3.1. Wort Preparation

The wort used in this study was obtained from the hydrolysis of a mixture of sugarcane juice (58 g L$^{-1}$) and corn (39% solids), as proposed by Douradinho et al. (2023) [6]. The syrup was obtained from concentrated sugarcane juice from a sugarcane mill located in Piracicaba, São Paulo, Brazil. The syrup had a soluble solids content of 69° BRIX and was then diluted to 58 g L$^{-1}$ fermentable sugars. The diluted sugarcane juice was clarified with lime and monobasic sodium phosphate, according to the procedure described by Sica et al. (2021) [7]. Corn was purchased from an agricultural store in the municipality of Piracicaba, near ESALQ. The corn grains were subjected to hammer milling to obtain fragments with a particle size smaller than 2 mm, using an automatic sieve.

The hydrolysis process began with the heating of the corn particulate suspension. For this, 639 g of corn particulate was added for each liter of sugarcane syrup with a concentration of 58 g L$^{-1}$ of total reducing sugars and a pH of 5.8. Then, following the hydrolysis steps proposed in [6], the mixture was preheated to a temperature of 55·°C and already contained 80 mg of the Liquozyme® (Novozymes, Copenhagen, Denmark) α-amylase enzyme (EC3.2.1.1). It was heated for approximately 40 min until the temperature stabilized at 88 °C, when an additional 80 mg of the same α-amylase enzyme Liquozyme® was added, and the mixture was kept under constant agitation for 150 min (liquefaction). After the corn starch liquefaction was complete, the system was cooled until the temperature stabilized at 65 °C. At this point, the pH of the mixture was adjusted to 5.0. Under these conditions, 224 mg of the Spirizyme® (Novozymes, Copenhagen, Denmark) glucoamylase enzyme (EC3.2.1.3) was added, and the mixture was kept under constant rotation for 150 min (saccharification) [6].

After the saccharification step, the resulting mixture, consisting of corn particulate and a sugary solution, was centrifuged using a Thermo Scientific® (Waltham, MA, USA) Sorvall

ST40R horizontal centrifuge at a rotation speed of 10,000 rpm (3924× *g*), at a temperature of 5 °C, for 10 min to separate the components. The sugar-rich solution obtained as the supernatant from the centrifugation was then filtered through a 210 μm sieve and used for the preparation of the wort, which was enriched with 600 mg L$^{-1}$ of urea. The addition of sugarcane juice (58 g L$^{-1}$ fermentable sugars) to corn (39% solids, w/v) during the hydrolysis process resulted in a wort with a total fermentable sugar content of 342.4 g L$^{-1}$. An overview of the experimental setup is shown in Figure 2. More details on the wort composition are given in Table 5.

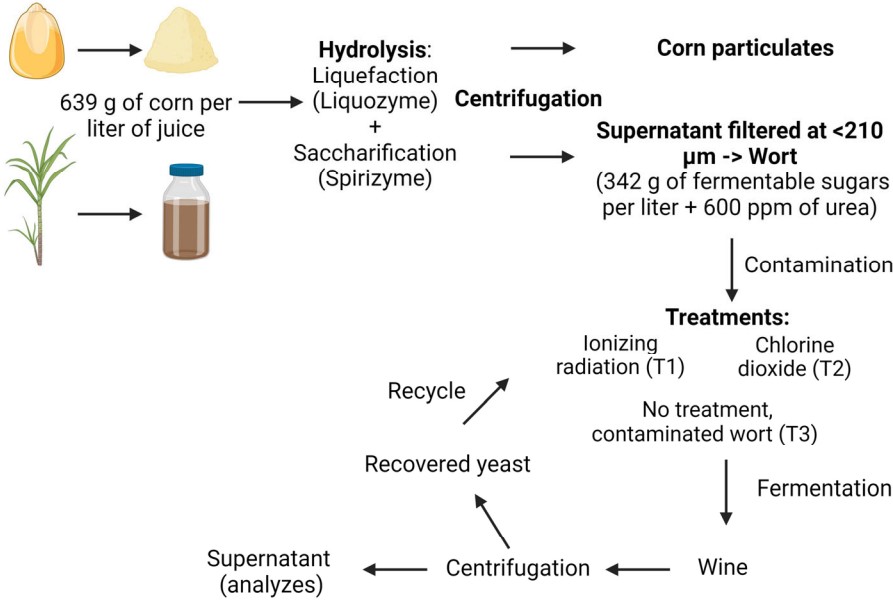

**Figure 2.** Diagram summarizing the hydrolysis process and the fermentation yeast recycling performed for wort disinfected with electron beam (radiation), wort disinfected with chlorine dioxide (ClO$_2$), and non-treated wort.

### 3.2. Wort Contamination and Disinfection Treatments

The bacterial contamination is mainly caused by the presence of mineral impurities (from soil), which are carried along with the raw materials supplied to the industry [12,24]. Therefore, we intentionally introduced 10 g L$^{-1}$ of the mineral impurities present in the raw materials into the wort, which was shaken for 12 h at 30 °C and was used as a real microbial contaminant typically found in the industrial processes [9,11,74]. The wort reached an average count of total mesophiles of 7.1 Log CFU + 1 mL$^{-1}$, bacterial count of 6.9 Log CFU + 1 mL$^{-1}$, total acidity of 2.7 g L$^{-1}$, and pH 4.9.

The contaminated wort was then divided into three equal portions. The first portion was irradiated with an electron beam (20 kGy) to sterilize the wort and create the T1 treatment. Another portion of the wort was not irradiated (0 kGy) but was supplemented with 125 mg kg$^{-1}$ of the antimicrobial chlorine dioxide (T2 treatment, positive control). The third portion remained untreated, with no irradiation or biocide addition (T3 treatment, negative control).

The wort irradiation was conducted using the electron accelerator of the Technological Radiation Center (CTR), at the Institute of Energy and Nuclear Research—IPEN/USP. The equipment is the industrial electron accelerator Dynamitron DC1500/25/4—JOB188 (Radiation Dynamics Inc., Edgewood, NY, USA), which operates with a power of 150 kW, a beam energy of 1.5 MeV, an electric current of 5.61 mA, and a beam width of 1.12 m. It emits an electron beam with a dose rate of 19.99 kGy s$^{-1}$ (19.99 kJ kg$^{-1}$ s$^{-1}$) [75]. The electron accelerator is a safe "on/off" radiation source that establishes a high voltage potential between a cathode and an anode in a vacuum tube to accelerate electrons and emit the electron beam. The energy transferred to the irradiated medium can randomly

remove electrons from the targeted atoms and result in a loss of the chemical identity of the molecules (inactivation) [76].

The wort was conditioned in glass containers in such a way that the wort layer had a thickness of 4 mm, which is a depth that the electron beam is able to penetrate. The materials were irradiated in a dynamic mode, with the aid of a treadmill that moved the containers at a speed of 6.72 m min$^{-1}$ 4 times under the 1.12 m beam width. The total radiation dose was selected based on [6]. For the treatment involving the biocide (T2), the chlorine dioxide dosage was consistent with the recommendations found in commercial formulations for microbial control during fermentation.

After the wort treatments, the entire process was carried out under aseptic and isolated conditions to prevent recontamination or treatment interference. After fermentation, the wine was centrifuged to recover yeast cells for use in subsequent fermentation cycles (yeast recycling). These fermentations were carried out in five consecutive and uninterrupted cycles, characterizing them as Melle-Boinot fermentations. The resulting wines were subsequently distilled and subjected to further biological and chemical analyses.

### 3.3. Analyses

3.3.1. Bacterial and Total Mesophilic Counts

Bacterial and total mesophilic counts were conducted using the serial dilution and pour plate technique on Plate Count Agar (PCA) medium. The plating procedures followed the protocols described by Oliveira et al. (1996) [77]. Serial dilutions were prepared in test tubes containing 9 mL of deionized water supplemented with 0.1% (*w/v*) peptone and sterilized. Petri dishes were incubated in a Marconi® oven (model MA415) at $30 \pm 0.5°$ C for 48 h to allow microbial growth, followed by colony counting. In counts aimed at determining the number of bacterial colonies exclusively, the fungicide cycloheximide (Actidione®, from Sigma-Aldrich, located in Darmstadt, Germany) was added to the culture media at a concentration of 10 mg L$^{-1}$ to inhibit yeast growth in order to count the total viable bacteria.

3.3.2. Yeast Viability

Yeast viability was assessed via differential staining of living cells with 0.1% methylene blue solution. Dead, live, and viable cells and buds were counted in a Neubauer chamber under a light microscope, as previously described by Douradinho et al. (2023) [6].

3.3.3. Ethanol Content

To determine the ethanol content, 25 mL of the supernatant from centrifuged samples was collected from each experimental unit at the end of fermentation. These samples were distilled using an MA 012/1 microdistillation apparatus (Marconi, Piracicaba, Sao Paulo, Brazil). After distillation, the ethanol content analysis was performed using the Schmidt Haensch Digital Densimeter EDM 4000, following the methodology described in [6,7,77,78].

3.3.4. Total Acidity

The total acidity of the centrifuged wine was determined according to the method proposed by Amerine and Ough (1981) [79]. For this purpose, 20 mL of a homogenized sample was transferred to an Erlenmeyer flask, to which 50 mL of deionized water and 3 drops of 1% phenolphthalein indicator solution (*w/v*) were added, and then the solution was titrated to pH 8.2. The total acidity was determined from the consumed volume of 0.1 N NaOH, and the result was expressed in grams of acetic acid per liter.

3.3.5. Fermentable Sugar Content

The content of fermentable sugars glucose, fructose, and sucrose was determined using the 3,5-dinitrosalicylic acid method as described by Miller (1959) [80]. The reaction solutions were prepared by adding 0.8 mL of a sample of the diluted must (25 times) and 0.2 mL of a 6 M hydrochloric acid solution in test tubes. These tubes were then heated for

7 min using a water bath system at 60 °C. The reaction was stopped by adding 1 mL of 2.4 M sodium hydroxide. The colorimetric reaction began with the addition of 2.0 mL of the 3,5-dinitrosalicylic acid reagent, followed by heating in boiling water (water bath) at 100 °C for 5 min. Absorbance was measured using a spectrophotometer at a wavelength of 546 nm.

### 3.3.6. Total Nitrogen

Nitrogen analysis was performed using a total organic carbon (TOC) analyzer (TOC-LT, from Shimadzu®, located in Tokyo, Japan). The calibration range for total nitrogen (TN) measurements spanned from 0.5 to 1000 mg $L^{-1}$. Sodium nitrate served as the calibration standard for TN measurements. Sample preparation involved dilution to a 1:1000 ratio using volumetric flasks filled with ultrapure water. After dilution, the samples were filtered through a 0.22 μm filter, and their pH levels were adjusted to the range of 2–3 via the addition of a small quantity of a 37% m $v^{-1}$ sodium chloride solution.

### 3.3.7. Minerals

The quantification of other elements, namely copper, cobalt, magnesium, manganese, phosphorus, potassium, sodium, and zinc, was carried out using inductively coupled plasma optical emission spectrometry (ICP-OES). For analysis, wort samples were subjected to wet oxidation digestion. The methodology used was adapted from the method compendium associated with the MARS 6 microwave system [81]. The digestion process involved the use of nitric acid and hydrogen peroxide until the samples were colorless and particle-free. The resulting solution was diluted with water to a final volume of 50 mL. The prepared solution was then filtered through a 0.22 μm filter and subjected to analysis. Commercial standards at a concentration of 1000 mg $L^{-1}$ (Specsol) were appropriately diluted with ultrapure water to generate the calibration curves required for the ICP-OES analyses, which ranged from 0.01 to 10 mg $L^{-1}$ for the elements of interest.

### 3.3.8. Ethanol Productivity and Fermentation Efficiency

Ethanol productivity was calculated as the amount of ethanol produced (in grams) per liter of wort per hour of fermentation. Fermentation efficiency was defined as the percentage ratio between practical and theoretical yields, based on the stoichiometric ethanol production per 100 g of fermentable sugars, according to Gay-Lussac (51.11 g of ethanol/100 g of sugars), as described by Douradinho et al. (2023) [6].

### 3.4. Statistical Analyses

The obtained results were subjected to analysis of variance (ANOVA) to test for significant differences among treatments. The Tukey test was used to compare means between treatments ($p < 0.05$). The experiment was performed in five randomized time blocks. Each of them had five fermentation cycles. The experimental design was a randomized block design with three treatments and five replicates. Treatments were compared within each cycle, and the cycles were compared within each treatment. These statistical analyses were performed using the SAS 9.4v software package.

## 4. Conclusions

Ionizing radiation can be highlighted as a promising treatment for microbial contamination control and the improvement of fermentation parameters. The results showed the inactivation of contaminants under the effect of electron beam irradiation, which led to an increase in the productivity, yield, and efficiency of fermentation by 0.21 g $L^{-1}$ $h^{-1}$, 2.6%, and 4.7%, respectively. Remarkably, the yeast maintained considerably high fermentation efficiencies even after undergoing five cycles. Furthermore, we observed that contamination levels remained consistently insignificant in the fermentation environment, creating the potential to eliminate the yeast acid treatment typically employed. This not only shortens the yeast recycling time, which usually takes two hours and requires the use of sulfuric acid,

but also reduces the need for external inputs and minimizes the requirement for antibiotics to control bacterial contamination.

The integration of this treatment holds great promise for facilitating the adoption of yeast recycling practices in corn-based ethanol production, whether used individually or in combination with sugarcane. The electron beam is proving to be a safe source of ionizing radiation to inactivate microbial contaminants, with no radioisotopes or radioactive waste, reducing or eliminating the need for antibiotics—resulting in no antibiotic residues on DDG or WDG (for animal feed) and no antibiotic-resistant strains of bacteria in the environment.

**Author Contributions:** Conceptualization, R.D., P.S., V.A., S.C. and A.B.; methodology, R.D. and A.B.; software, R.D., S.P., P.S., D.P., V.G. and A.B.; validation, R.D., S.P., L.A., J.M.A.R.d.A., S.C. and A.B.; formal analysis, R.D., P.S., D.P., V.G., S.P., S.C. and A.B.; investigation, R.D., M.O., A.U.P., L.M. and E.M.; resources, J.M.A.R.d.A., S.P., L.A., V.A., S.C. and A.B.; data curation, R.D., P.S., S.P. and A.B.; writing—original draft preparation, R.D. and P.S.; writing—review and editing, R.D., P.S., S.C. and A.B.; visualization, R.D., P.S., J.M.A.R.d.A., S.P., L.A., V.A., S.C. and A.B.; supervision, R.D. and A.B.; project administration, R.D. and A.B.; funding acquisition, R.D., P.S., J.M.A.R.d.A., S.P., L.A., V.A., S.C. and A.B. All authors have read and agreed to the published version of the manuscript.

**Funding:** This research was funded in part by "Coordenação de Aperfeiçoamento de Pessoal de Nível Superior—Brasil" (CAPES), grant number 001.

**Data Availability Statement:** The authors will make the data available, if necessary.

**Acknowledgments:** We gratefully acknowledge the support of Research Centre for Greenhouse Gas Innovation (RCGI), hosted by the University of São Paulo (USP) and sponsored by Sao Paulo Research Foundation (FAPESP) (2020/15230-5; 2014/50279-4) and sponsors, and the strategic importance of the support given by ANP (Brazil's National Oil, Natural Gas and Biofuels Agency) through the R&DI levy regulation.

**Conflicts of Interest:** The authors declare no conflicts of interest.

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
