# Peer review of "Assessing Ionizing Radiation and Chlorine Dioxide (ClO2) as Potential Aseptization Treatments for Yeast Recycling on Mixed Wort of Corn and Sugarcane in Brazil"

_stresses, doi:10.3390/stresses4010009_

Round 1

Reviewer 1 Report

Comments and Suggestions for Authors

The manuscript assess ionizing radiation and chlorine dioxide (ClO2) as 2 potential aseptization treatments for yeast recycling on mixed wort of corn and sugarcane in Brazil. The manuscript could be published after correction.
1. Introduction
- Line 53: 
produce compounds. Explain further the compounds includes examples
- Line 58: 
antibiotics are used to control these contaminating bacteria. Is there any other methods to control these contaminating bacteria? Compare and contrast with other literature. Report few examples of study and their main findings.
- Line 65: Why authors
propose the adoption of the Melle-Boinot yeast recycling process? Justify with citation.
- Overall,
need some improvements on the writing and scoping the problem statement with national/international agenda such as agrofood policy, Sustainable Development Goals 2030, etc.
2. Materials and method
- Why materials and method were placed after results?
- Line 346: the first semester of 2022. it is suggested to report the time duration in terms of which month
- Section 4.3: it is suggested to divide this section to few sub-section of analysis i.e. 4.3.1 Yeast viability....4.3.2...... etc
3. Results and Discussion
- It is suggested to combine both sections.
- Line 160: 
the treatment significantly reduced the yield compared to the negative control and exhibited lower productivity and efficiency compared to ionizing radiation. Explain the mechanism/how the treatment and ionizing radiation in details
- Line 176: 
Iizuka, Shibabe, and Ito (1969) outdated reference
-Line 199: 
Gutierrez et al. (1990) outdated reference
-Line 205: 
Heerde & Radler (1978) outdated reference
-Line 216: 
Oliva-Neto and Yokoya (1994) outdated reference and many more. Re-check and replace with updated references. More references within 5 years time (2024, 2023, 2022, 2021, 2020)
- Line 276: 
combining energy cane juice with corn also improved fermentation efficiency compared to using only corn. How? Explain
4. Conclusion
- Report main findings with numerical value in conclusion.

Author Response

The comments are in the word file.

Reviewer 2 Report

Comments and Suggestions for Authors

Minor revision. 

1. Please compare radiation approaches for different fermentation types in methodology or introduction for example with 10.1016/j.ijhydene.2019.03.216

2. Why do You choose Anova in statistics please add reasons?

3. Please describe more precisely the source of irradiation with the type of irradiation used in the research.

4. For better readership of people unfamiliar with methods please state using proper safety rules.

Author Response

The comments are in the word file.

Reviewer 3 Report

Comments and Suggestions for Authors

Remark to Abstract: The abstract is written in poor English and should be corrected and edited, e.g., as follows:

Yeast recycling, which is a common practice in sugarcane ethanol plants, could be expanded if it could be successfully implemented in corn-based ethanol production. However, the recycling process of the yeast remaining after the fermentation step is hampered by contaminating microorganisms, which reduce the efficiency of fermentation and destroy the fermentable sugars. Currently, these microorganisms are destroyed using various biocides. Our study examined electron beam irradiation as an alternative method to completely inactivate contaminants and minimize their effect on recycled yeast. For this purpose, we compared the sterilization of the wort by electron radiation (20 kGy) and the treatment by chemical biocide such as chlorine dioxide (125 mg/kg) with non-treated wort. We carried out five fermentation cycles using fed-batch systems containing 300 g L-1 of fermentable sugars. Results showed the inactivation of contaminants under the effect of the electron beam irradiation, which led to an increase in productivity, yield, and efficiency of fermentation by 0.21 g L-1h-1, 2.6% and 4.7%, respectively. Thus, irradiation of contaminated wort with an electron beam may be a suitable alternative to chemical biocides and would allow the use of recycled yeast in corn-based ethanol plants.

All Tables.  Remark: The authors should calculate and add the average results of all five experiments to these tables.

Remarks to section 4 “Materials and Methods”: (1). In this section, the authors indicated that the dose of irradiation was 20 kGy, while the dose rate was 19.99 kGy s-1. Does this mean that the irradiation time was only 1 second? (2). The authors should note, what thickness (depth) of the wort was irradiated, whether static or dynamic irradiation mode was used, etc.

Additional remark:

For industrial use of the proposed method, the author must provide calculations of the economic efficiency of using irradiation for sterilization in comparison with biological and chemical sterilization methods. For example, if using ClO2 having a price of $2/kg ClO2 and a dose of 125 mg per kg wort, then the expense for sterilization of 1 ton wort will be $ 0.25. On the other hand, the irradiation of the 1 ton wort by the dose of 20 kGy requires 5.56 kWh energy at minimum (without accelerator price and irradiation efficiency), i.e., about $1. Thus, sterilization with irradiation is at least 4 times more expensive than treatment with ClO2.

Comments on the Quality of English Language

English should be improved

Author Response

The comments are in the word file.

Round 2

Reviewer 3 Report

Comments and Suggestions for Authors

The authors revised the manuscript taking into consideration the remarks of the reviewer. However, the revised version still contains some shortcomings that need to be corrected.

Page 6. Line 2. Remark: This sentence should be supplemented as follows, “In this study, chlorine dioxide with a dosage of 125 mg L-1 was added directly......

Page 6. Lines 8-10. “As a strong oxidant that has antimicrobial properties, the ClO2 acted against the microrganisms during the fermentation process undesirably affecting the yeast performance.  Remark: This sentence is unclear and contains grammatical errors; therefore, it should be corrected, as follows, “Being a strong oxidizing agent, ClO2 had an undesirable effect on yeast performance”.

Page 6. Lines 10-12. “While the ionizing power of the electron beam to sterilize the wort was immediately (~1 second) expressed when the electrons shot the material before fermentation, with no damage to the yeast.  Remark: This sentence should be corrected, as follows, “The ionizing ability of the electron beam to sterilize wort was immediate (~1 second) and without damaging the yeast”. In addition, this sentence is in the wrong place (Section 2.1.1.) and should be moved to Section 2.1.2. Ionizing radiation”, after “and improving fermentation parameters. (Line 35).

Page 6. Lines 14-18. “Thus, even the biocide treatment is cheaper (~US$ 3 m-3) than the costs with the 270 kWh energy demand of the 20 kGy electron accelerator (US$ 4 m-3) [33,34], in the conditions of this study, the chlorine dioxide treatment did not show promising on controlling microbial contamination and did not improve fermentative parameters compared to the negative control. Remark: This sentence needs to be corrected as follows, “Thus, although ClO2 biocide treatment of the wort with an amount of 125 mg L-1 is cheaper (~US$ 3 m-3) than the energy cost of an electron accelerator to produce a dose of 20 kGy ($4 m-3) [33,34], this study showed that ClO2 treatment was less effective at suppressing microbial contamination than electron irradiation”.

Comments on the Quality of English Language

English needs to be checked and corrected

Author Response

The comments are in the word file.
